# Pavement Distress Estimation via Signal on Graph Processing

**DOI:** 10.3390/s22239183

**Published:** 2022-11-25

**Authors:** Salvatore Bruno, Stefania Colonnese, Gaetano Scarano, Giulia Del Serrone, Giuseppe Loprencipe

**Affiliations:** 1Department of Civil, Constructional and Environmental Engineering, Sapienza University of Rome, Via Eudossiana 18, 00184 Rome, Italy; 2Department of Information Engineering, Electronics and Telecommunications, Sapienza University of Rome, Via Eudossiana 18, 00184 Rome, Italy

**Keywords:** pavement distress detection, pavement condition index, pavement management program, signal on graph processing, automated distress evaluation systems, Bayesian estimator

## Abstract

A comprehensive representation of the road pavement state of health is of great interest. In recent years, automated data collection and processing technology has been used for pavement inspection. In this paper, a new signal on graph (SoG) model of road pavement distresses is presented with the aim of improving automatic pavement distress detection systems. A novel nonlinear Bayesian estimator in recovering distress metrics is also derived. The performance of the methodology was evaluated on a large dataset of pavement distress values collected in field tests conducted in Kazakhstan. The application of the proposed methodology is effective in recovering acquisition errors, improving road failure detection. Moreover, the output of the Bayesian estimator can be used to identify sections where the measurement acquired by the 3D laser technology is unreliable. Therefore, the presented model could be used to schedule road section maintenance in a better way.

## 1. Introduction

### 1.1. Background

The accurate and reliable assessment of road surface degradation is very important for planning appropriate maintenance strategies, primarily to ensure safety and comfort of the road users. Pavement management systems are the tools traditionally used to assist road managers in decision-making regarding pavement maintenance and rehabilitation; they are mainly based on distress data analysis [1]. According to the World Bank, in fact, the pavement condition of a country’s road network determines its economic level [2]. To regulate the activities required to maintain adequate pavement performance, it is essential to gather information on the state of health of the road surface: in this regard, the data can be collected by visual inspections or by automated detection solutions [3].

Visual deterioration inspections have been the most common method used to collect road data for many years: qualified personnel analyze the pavement conditions by walking or traveling in a car at slow speeds [4]. Traditionally, the pavement condition index (PCI) is used to describe the general condition of a pavement section based on road data collected. PCI numerical values, between 0 and 100, allow the road manager to capture useful information about the most suitable intervention techniques to adopt [5]. The ASTM D6433 [6] and ASTM D5340 [7] formalized and standardized the PCI calculation methods. These standards are based on detecting the pavement distress types encountered at the road surface and then on distinguishing each of them by different severity levels. However, the visual surveys are time-expensive, wasteful, and subjective [8].

Therefore, in recent years, automated survey methods have increasingly been employed [9] through innovative and low-cost technologies [10] for the analysis of road pavements, evaluating their potential to improve the automation and reliability of hazard detection. These automatic methods are safe, fast and repeatable, at the expense of the high cost of the acquisition equipment and the long processing and calculation times [11]. Nondestructive methods of diagnosing the decay of a road surface can include the use of ground-penetrating radar (GPR), laser scanning technology and finite element method (FEM) calculations to evaluate the factors that contribute to early pavement cracks [12].

Most of these methods are pavement crack detection systems based on images [13]: for example, Mataei et al. proposed the structure from motion (SfM) technique in [14], applied on a road pavement, to accurately measure the pavement texture. Du et al. [15] processed the conditions of the road surface, automatically detecting and measuring road discomfort from images based on unmanned aerial vehicles (UAVs). A similar approach was used by Garilli et al. in [16] to detect natural stone pavements, analyzing two supervised classification approaches: the semi-automatic classification plugin for QGIS and a convolutional neural network (CNN). An approach for the automatic detection of raveling [17] was also constructed from image sample processing. A stochastic gradient descent logistic regression (SGD-LR) was subsequently implemented to classify the image samples into two categories of nonraveling and raveling based on a set of extracted characteristics. In the studies [18,19], low-cost data acquisition systems were developed. The systems rely on smartphones equipped with an accelerometer and a geographical positioning system, in one case, and a Sharp IR-based sensor and an accelerometer, in the other, to measure the pavement roughness. To overcome differences coming from dissimilar smartphone sensor properties, an algorithm that uses threshold-based and machine learning approaches for the near real-time detection and the classification of road surface anomalies was proposed in a recent study [20]. Furthermore, the identification, representation, and surface quantification of newly formed holes are important for timely maintenance and repair. This study [21] presents a low-cost based approach for the detection of potholes on asphalt road pavements in urban areas from 2D images using the fuzzy c-means algorithm (FCM). Analyzing the pavement data obtained from 3D laser scanning, meanwhile, an automatic defect detection method was proposed in [22] to simultaneously detect typical cracks and information on deformation defects. Dan and Dan discuss an intelligent approach to the automatic recognition of pavement cracks in [23] based on a two-dimensional amplitude and phase estimation method able to filter low-frequency information. Li et al. proposed the interleaved low-rank group convolution hybrid deep network (ILGCHDN) in [24], with the aim of recognizing cracks and non-cracks on complex road surfaces.

However, it has to be noticed that the output of any data collection system includes errors [25] that may depend on several factors related either to the instrument used to make the measurement or to the personnel performing the action of measuring: researchers around the world have adopted many methods and algorithms to overcome these limits. Dong et al. in [26] summarized and discussed more than 40 data analysis methods useful for investigating raw data coming from pavement detections, including statistical tests, experimental design, regressions, count data model, survival analysis, stochastic process models, supervised learnings, unsupervised learnings, reinforcement learnings, and Bayesian analysis applied in pavement engineering. It can be said that the accurate and reliable estimation of road surfaces is still an open challenge.

### 1.2. Objectives of the Current Work and Problem Statement

Advanced pavement data collection vehicles (Figure 1), professional range of equipment, represent a modern technique to monitor the road surface conditions over time [27].

In this study, an automated pavement condition data collection using laser crack measurement system (LCMS) [28] was used to identify the distresses detected in some field tests carried out in Kazakhstan [29]. The system works properly under various light conditions (both in sunny and shaded areas), and under low-contrast-intensity conditions, at a driving speed up to 100 km/h [30]. Specifically, it consists of two units, each of which combines a high-power spread line laser with a facing-down high-speed camera in an off-axis configuration, as shown in Figure 2:

When combined, the two 3D laser units project a 4 m wide laser line, and its image is captured by the cameras [31]. The pavement data are collected under 1 mm transverse resolution and a depth accuracy of 0.5 mm [32]. A detailed modeling of the 3D road surface is acquired as the surveying vehicle travels along the road using a signal from an odometer to synchronize the sensor acquisition [33]. Based on the output, data post-processing (using specific algorithms within the detection system data processing software) makes it possible, in a single run, to automatically identify the type, severity, and extension of the surveyed distresses [34]. Figure 3 illustrates an example image of a pavement with defects automatically overlaid.

However, experimental tests have proved that only cracks equal to and greater than 2 mm wide can be effectively detected using 3D laser technology [30]. As a further confirmation of the above, the accuracy of block crack or irregular crack identification has been found better than that of transverse and longitudinal cracks: the latter ones are generally narrower [35]. As for the detection of the transverse cracks, it is less accurate than that of the longitudinal cracks; it is due to the resolution of the laser scanning data, which is higher in the transverse direction than that in the longitudinal direction [22]. Laefer et al. in [36] noted that using laser scanner technology, as the crack width increases, the detection of the crack depth is more accurate. In addition, the accuracy in detecting the crack depth increases as the scanning angle (the offset angle between the crack and the laser scanner) decreases, while it decreases as the scanning distance (the distance between the crack and the laser scanner) increases. In other studies [37], the 3D systems incorrectly reported the joint between the kerb and the pavement as a crack. Detecting sealed cracks is also not easy using LCMS sensors, especially as the sealed crack ages and the sealant wears away and chipping begins [38]. In some field tests on airfield pavements in Ireland [39], a check on the automated distress data outputs showed some problems with the identification of short lengths of hairline longitudinal and transverse cracking and short lengths of low-severity sealed cracks and with the initial misdiagnosis of transverse and diagonal timing as linear cracking. Using deviation analysis of LCMS measurements from manual measurements, Williams et al. in [40] detected some very severe in situ transverse cracks as potholes; moreover, most of the cracking manually measured as longitudinal was reported as meandering/block cracking.

Therefore, the main objective of this study is to solve these limitations and to try to achieve high accuracy in reconstructing noisy or missing real data: a Bayesian estimator of signal on graph has been derived [41].

In particular, we formulate the problem of distress correction as the estimation of a suitably defined signal on graph (SoG), and we propose a novel methodology aimed at correcting possibly altered distress measurements based on the adopted SoG model. The main contributions of the paper are as follows:We introduce a novel signal on graph model of the observed distress metrics; specifically, we consider the distress metrics acquired at different spatial sections as signal values associated with the vertices of a graph. The graph edges represent (i) the correlation among different distress metrics at a given spatial section and (ii) the correlation among spatially adjacent measurements of the same metric.We derive a novel optimal estimator of the distress metrics. The novel nonlinear Bayesian estimator is built on top of a recent model for signal defined on graphs [42], and it is optimal in the sense that it minimizes the mean square estimation error. In pavement engineering, the Bayesian analysis can be used to obtain and update the values of parameters based on the historical data and to provide a posterior probability distribution for the parameters [43].

Finally, the PCIs have been calculated based on both raw and processed data in order to compare them and check the accuracy of the Bayesian estimator.

A framework for the pavement distress estimation via SoG processing is shown in Figure 4.

The introduced signal on graph model paves the way for further studies, such as the assessment of the measurement quality as well as the distress feature extraction for deep learning strategies [44]. 

## 2. Measurement and Evaluation of Pavement Conditions

The analysis of pavement conditions is a fundamental phase in defining an adequate infrastructure maintenance program, optimizing the budget allocation [45]. The deterioration of a road pavement can be divided into functional and structural characteristics [46]. Functional means that the structure is still efficient, but friction or roughness are compromised, so as to make the movement of vehicles unsafe and uncomfortable; structural, instead, indicates that the pavement shows top-down cracking [47] or other damages (potholes, ruts, etc.) due to repeated traffic loads.

The pavement condition index (PCI) quantifies the current conditions of the pavement considering the type, extent and severity of the detected deterioration in an objective, rational and scientific way. The index is useful for assessing the rate of deterioration, establishing a priority scale of the main maintenance and rehabilitation activities; its calculation procedure is explained in the Standard ASTM D6433 [6] and ASTM D5340 [7], as said in the previous section of this paper. Then, it is a priority to characterize, in terms of both type and severity, the set of distresses on the pavement surface.

The pavement surface degradations analyzed in this paper—among those listed in the standards [6,7]—are those acquired and identified by the LCMS in the field tests: they can be divided into cracks and plastic or viscous surface deformations. Table 1 lists the road surface defects taken into account in the model proposed in this paper; the corresponding descriptions are provided below:Alligator cracking occurs in areas subjected to repeated traffic loads. It originates at the bottom of the hot mix asphalt (HMA) layers, where the tensile stress–strain induced by loads is highest. The cracks propagate on to the surface in parallel ways at first, then interconnect to form polyhedral signs at acute angles, smaller than 60 cm on the longest side;Rutting is a road surface depression or groove due to the travel of wheels or skis. The deformation of the asphalt concrete pavement or subbase material can cause the ruts, which can become unsafe during rainy events, when the paths are filled with water;Longitudinal and transverse cracking happens lengthwise or crosswise at approximately right angles to the pavement’s centerline or laydown direction. These types of cracks are not usually load-associated but can occur, for example, at low temperature or because of asphalt hardening. Nonfilled crack widths less than 6 mm for airport pavement and 10 mm for road pavement are always considered of low severity.Block cracks divide the pavement surface into rectangular pieces whose extensions generally range between 0.1 m^2^ and 10 m^2^. They are caused by the shrinkage of the asphalt concrete due to the cyclical variation of daily temperature. Whatever is the load phenomenon, the asphalt manifests an excessive hardening, and so it can occur on non-trafficked sections;Raveling is due to the wearing away of the pavement surface with the loss of bituminous material and the consequent dislodging of the aggregate materials. The phenomenon indicates poor quality of the mixture and a hardening of the asphalt binder.Lane/shoulder drop-off is the difference in elevation between the traffic lane and the shoulder. Generally, the shoulder settles because of the consolidation or pumping of the underlying material.Potholes are bowl-shaped depressions with a diameter less than 1 m on the road surface. The edges are generally sharp, and the sides are vertical. They are formed when small portions of pavement are removed by traffic, and the phenomenon is accelerated by the presence of water that can stagnate inside them.

In this table, the ID number identifies the numbering that will recall the single defects in this paper. The absence in Table 1 of block cracking with a low degree of severity is justified by the lack of data on this metric in the totality of the km investigated in the experimental phase.

Once the relevant defects are known, the procedure to calculate the PCI can be implemented [48]. The PCI of each road section is calculated using Equation (1), adding up the total quantity of each type of distress, with its own severity level and density coming from inspection data. The units for the quantities may be square meters, linear meters, or number of occurrences depending on the distress type. Let P denote the total number of distress types taken into account and mi denote the number of degrees of severity per the *i*-th type of distress:(1)$$PCI=100-\sum_{i=1}^{P}\sum_{j=1}^{m_{i}}\alpha \left[T_{i},S_{j},D_{ij}\right]\cdot F\left(t,d\right)$$
where *α* is the deduct weighting value depending on distress type *T_i_*, level of severity *S_j_*, and density of distress *D_ij_*; *i* = 1, …, *P*, *j* = 1, …, *m_i_* and *F*(*t*,*d*) is the adjustment factor for multiple distresses, which varies with the total summed deduct value *t* and number of deducts *d*.

The PCI of the pavement condition is a numerical value between 0 and 100, where 0 is the worst possible condition and 100 is the best possible condition, as can be observed in Figure 5.

## 3. Proposed Model of Road Distress Estimation as a Signal on Graph

Herein, we model the automated distress data from the LCMS as signal values measured at the vertices of a graph. Before turning to mathematics, let us outline the association between the road distress measurements and the graph structure. To this aim, let us consider a section in a road lane. For each section (1, 2, …,) usually 100 m long and for a total of M sections, a set P of distress metrics (a, b, c, …, n) is measured by the LCMS. This is exemplified in Figure 6. For each section, distinguished by a different color, we consider as many graph nodes as the number of different measured metrics. The distress metrics are assumed to be correlated to each other. In the section related to the experimental results, we present the results of a preliminary co-occurrence analysis of collected distress data in a suitable measurement set, and we show few metrics to correlate to each other. We represent the correlation between different metrics taken at the same section links between corresponding nodes. Finally, spatially adjacent nodes relative to the same metric are deemed connected, too. This is illustrated in the lower part of Figure 6, where we recognize the constraint graph structures. The total number of nodes is given by the number M of inspected sections times the number P of distresses evaluated by the automated system.

For a useful interpretation, we assume a few metrics to be correlated to each other. The correlation, estimated by means of a preliminary co-occurrence analysis of collected distress data, is represented by links between nodes relative to the same section; see for instance the pairs (1,b) and (1,d). In addition, nodes representing the same metric at adjacent spatial sections are connected as well; see for instance the pairs (1,b) and (2,b). The total number of nodes is given by the number M of inspected sections times the number P of distresses evaluated by the automated system.

With these positions, let us analytically introduce the SoG processing-based model adopted for metric estimation. Formally, let us introduce a graph *G* = (*V*, ***E***), where *V* denotes the set of *N* vertices and ***E*** the set of *N_E_ ≤ N^2^* edges (i.e., real weights) associated with the selected node pairs. The graph connectivity is represented by the *N*x*N* adjacency matrix **A**, whose *i*,*j* element represents the weight of the edge between the *i*-th and *j*-th vertices, as well as by the Laplacian matrix **L** = **A − D**, **D** being the diagonal matrix of the graph node degrees.

The *i*-th vertex of the graph is associated to a signal ***x****_i_*, *i* = 0, …, *N* − 1, representing a value acquired at the *i*-th vertex. Graph *G* representing the metrics has a particular structure, illustrated in Figure 6 (bottom): The elementary graph connectivity representing the metric correlation at a given spatial section is replicated at different sections; moreover, the corresponding vertices representing the same metric at adjacent spatial sections are connected. This structure is the result of a product of graphs, as exemplified in Figure 7 for the toy case of P = 5 metrics measured at M = 4 investigated sections. The graph in Figure 7a represents the similarity between metrics at a given section, the path graph in Figure 7b represents the similarity between the same metric over adjacent spatial sections and Figure 7c represents the overall product graph. For a product graph *G*, the Laplacian matrix **L** is analytically found as the Kronecker sum of the Laplacian of the elementary graphs *G*_1_, *G*_2_ as: **L** = **L^(1)^** ⊕ **L^(2)^**.

For the case under concern, graph *G*_1_ models the similarity between the P metrics. The adjacency matrix coefficients $a_{ij}^{\left(1\right)}\;i,j=0,\;\ldots P-1$ are calculated according to Equation (2) as the correlation coefficients between the *i*-th and *j*-th distress metric values. namely
(2)$$a_{ij}^{\left(1\right)}=\frac{\sum_{n}y_{n}^{\left(i\right)}y_{n}^{\left(j\right)}}{\sqrt{\sum_{n}{\left(y_{n}^{\left(i\right)}\right)}^{2}}\sqrt{\sum_{n}{\left(y_{n}^{\left(j\right)}\right)}^{2}}}$$
where $y_{n}^{\left(i\right)}$ and $y_{n}^{\left(j\right)}$ are the possibly noisy measured values of the *i*-th and *j*-th distress metrics on n=0, …M−1 road sections, centered with respect to their mean and normalized to unitary variance. Graph *G*_2_ models the similarity between spatially adjacent samples of the same metric. The adjacency matrix coefficients $a_{ij}^{\left(2\right)}$ = 1 if |i−j|=1 and 0 otherwise. From a computational complexity point of view, let us observe that when the number P of measurements is large, the computation of the Kronecker sum is heavy, and it can be avoided by suitably managing the sparse adjacency matrices.

## 4. Nonlinear Bayesian Estimation of the Road Distress

Stemming from the above settings, we herein introduce the model of the noisy observed distress metrics in order to derive a Bayesian estimator of noisy or missing measurements. Bayesian estimation exploits a priori knowledge on the data to be recovered, typically in the form of probabilistic priors, and it is of paramount importance in real-world data modeling due to its ability to conjugate the prior statistical model with the information pertaining the observed data [49]. Herein, we derive the Bayesian estimator in closed form. Its computation does not require a training stage leveraging a large annotated dataset as it occurs in supervised machine learning techniques such as linear and nonlinear regression or decision trees [50,51,52]. On the contrary, the Bayesian estimator leverages few model parameters (e.g., data mean and variance) that can be straightforwardly estimated on the data. 

In the following, we introduce the main notation and the definition of the Bayesian estimator as the optimal nonlinear estimator from the point of view of the square of the error. Then, we derive its analytical formulation in closed form.

The set of P measurements acquired at M different road sections $y_{n}^{\left(i\right)}$, i=0, …P−1,  n=0, …M−1 is modeled as shown in Equation (3):(3)$$y_{n}^{\left(i\right)}=x_{n}^{\left(i\right)}+w_{n}^{\left(i\right)},\;\;\;i=0,\;\ldots P-1,\;n=0,\;\ldots M-1$$
where $x_{n}^{\left(i\right)}\;$ is the ground truth signal on graph representing the actual distress and $w_{n}^{\left(i\right)}$ is independent additive noise modeling the error of the acquisition system. Let us observe that the underlying graph model allows us to associate to each measurement $y_{n}^{\left(i\right)}$ a set $S_{n}^{\left(i\right)}$ of ν neighboring measurements, either referred to the same road section or to spatially adjacent ones. With these positions, we leverage the knowledge of the ν neighboring measurements in $S_{n}^{\left(i\right)}$ to improve the estimate the ground-truth value $x_{n}^{\left(i\right)}$. According to Bayesian estimation theory, the optimal minimum mean square error (MMSE) estimator of $x_{n}^{\left(i\right)}$, given the measured value $y_{n}^{\left(i\right)}$ and the ν neighboring measurements in $S_{n}^{\left(i\right)},$ is computed as the following expected value (Equation (4)):(4)$${\hat{x}}_{n}^{\left(i\right)}=E\left\{x_{n}^{\left(i\right)}|y_{n}^{\left(i\right)},\;y_{t},t=1,\cdots v;y_{t}\in S_{n}^{\left(i\right)}\right\}$$
where, without a loss of generality, we have compactly denoted the measurements in $S_{n}^{\left(i\right)}\;$ as $y_{t},t=1,\ldots \nu$.

In order to compute the above expectation, we model the true distress metric $x_{n}^{\left(i\right)}\;$ resorting to a recently introduced probabilistic model [42], which allows us to compactly account for the correlation between metrics measured at graph vertices. Stemming from the model in [42], herein we consider the following probability density function of the generic ground truth metric x conditioned to the ground-truth neighboring values $x_{i},i=1,\ldots \nu$ as defined in Equation (5):(5)$$p_{X|X_{1}\ldots X_{v}}\left(x|x_{1}\cdots x_{v}\right)=p_{0}\delta \left(x\right)+p_{1}Ke^{-\sum_{i=1}^{v}\beta_{i}{\left(x-x_{i}\right)}^{2}}$$
where we assume that with probability $p_{0}$, the observed distress metric is zero valued, and with probability $p_{1}=1-p_{0}$, it obeys to the Markovian signal on the graph model. The parameter K is a normalization factor and the parameters $\beta_{i},\;i=1,\ldots \nu$ weight the similarity of the current metric x with the neighboring ones $x_{i},i=1,\ldots \nu$. 

In the following, a few rules of thumb for setting the probability density function parameters will be illustrated. We assume the observation noise to be normally distributed, so that the observed metric $y_{n}^{\left(i\right)}$ is related to the ground truth metric $x_{n}^{\left(i\right)}$ by the following conditional density function (Equation (6)): (6)$$p_{Y|X}\left(y|x\right)=\frac{1}{\sqrt{2\pi }\sigma_{W}}e^{-\frac{{\left(y-x\right)}^{2}}{2\sigma_{W}^{2}}}$$

With these positions, the optimal minimum mean square error estimator of the considered metric, i.e., the expected value in (3), is derived in closed form by extending the algebraic approach adopted in [53] with the application of blind equalization and in [54] with the application of multichannel image deconvolution. For the sake of compactness, we omit the details here and come up with the final formula, which nicely blends the noisy measure y and the signal values in the neighbor set $S_{n}^{\left(i\right)}$ to provide the optimal estimate.

The optimal MMSE estimator $\hat{x}$ in (4) is computed as defined in Equation (7):(7)$$\hat{x}=\frac{\mu_{1}}{1+\frac{1}{K}\frac{p_{0}}{1-p_{0}}exp\left\{-\left(\mu_{2}-\mu_{1}^{2}\right)\sum_{i=1}^{v}\beta_{i}\right\}}$$
where
$$\mu_{1}=\frac{\beta_{0}y+\sum_{i=1}^{v}\beta_{i}x_{i}}{\sum_{i=1}^{v}\beta_{i}},\;\mu_{2}=\frac{\beta_{0}y^{2}+\sum_{i=1}^{v}\beta_{i}x_{i}^{2}}{\sum_{i=1}^{v}\beta_{i}}$$
represent the linear and quadratic weighted averages.

In Equation (7), we recognize that $\hat{x}$ is a nonlinear function of the noisy observation *y* and of its neighboring values. For given known values in the neighboring nodes, the Bayesian estimator $\hat{x}$ is a nonlinear function of *y*, namely $\hat{x}=\eta \left(y\right)$. The action of the nonlinearity η(y) is exemplified in Figure 8, which plots $\hat{x}$ versus *y* for assigned neighboring values and where we highlight the nonlinearity action for the case of a measured value *y* equal to 6.2 and observed neighboring values equal to 2.7, 2.9, 2.9, 3.3, and 3.4. In Figure 8a, we consider a constant value of the parameter $\beta_{0},$ and then we consider various values of $p_{0}$. The extent of the attenuation effect of the nonlinear estimator on small *y* values varies with the a priori probability $p_{0}$ of zero-valued distress metrics; in addition, the observation is corrected to result in line with the neighboring values. In Figure 8b, we consider a constant value for $p_{0}$ and various values of $\beta_{0}$, representing the quality of the measurement *y*. For a much-degraded observation, i.e., $\beta_{0}=0,$ the nonlinear estimator boils down to a constant determined by the neighboring values only. For increasing quality, the nonlinear estimator tends to leave the observation unaltered. 

In summary, the observed value, with a statistical nature of its own (it could be zero or related to its neighbors), is corrected according to the formulated aprioristic hypothesis, as expressed by the parameters of nonlinearity. In particular, the $\beta_{0}$ parameter represents the hypothesis of accuracy of the observed data: As $\beta_{0}$ tends to the unit value, we have higher confidence in the input data, and therefore, the lower is the correction made by the nonlinearity. As regards $p_{0}$, it provides information related to the expectation of having zero values in measurements: As $p_{0}$ tends to the unit value, the hypothesis that the ground truth value is equal to zero becomes prominent. 

In the following section, we apply the proposed signal on graph model and the associated Bayesian estimation procedure to real road distress measurements.

### 4.1. Performance of the Bayesian Estimator for the Toy Case

To this aim, we consider the graph in Figure 7c, and we associate with each of the P = 5 metrics a set of M = 4 sections.

Herein we illustrate the performance achieved by the Bayesian estimator on road data when they are affected by additive acquisition noise. Firstly, we consider the graph in Figure 7c, modeling the acquisition of P = 5 metrics over M = 4 sections $x_{n}^{\left(i\right)}\;\left(i=0,\ldots P-1,\;n=0,\ldots M-1\right)$ selected by the road dataset considered in this research work. Then, the observation $y_{n}^{\left(i\right)}$ is obtained from $x_{n}^{\left(i\right)}$ by adding white noise; for concreteness sake, we consider normally distributed noise, typically used to model the superposition of different random contributions [55], with signal to noise ratio SNR= 5 dB. Therefore, we compute the above-described Bayesian estimator ${\hat{x}}_{n}^{\left(i\right)}$ to each graph vertex, obtaining the set of restored values.

To assess the ability of the Bayesian estimator in recovering the data, we compute the achieved mean square error (MSE) defined as follows (Equation (8)):(8)$$MSE=\frac{1}{MP}\underset{n}{\sum }\underset{i}{\sum }{\left(x_{n}^{\left(i\right)}-{\hat{x}}_{n}^{\left(i\right)}\right)}^{2}$$

We average the MSE over 10 Montecarlo runs and plot the result in Figure 9 (“Bay” bar). For comparison sake, we consider different linear and nonlinear estimators [56,57], namely the mean and median of spatially adjacent measurements (“Mean”, “Median”), which are optimal in that they minimize the mean square or the mean absolute error for suitable kinds of noisy measurements. In principle, also the successive convex approximation (“SCA”) signal estimator in [42], which minimizes the mean square error of the estimate subject to a smoothness regularization constraint, could be considered. Still, since the SCA estimator poses a computational challenge over huge datasets such as the real roads datasets considered in this study, we resort to an over-regularized version of the SCA estimator, which is obtained by averaging the observations over the neighboring values over the considered graph (“Graph”). The abovementioned methods do not require large human annotated training datasets but only a global statistical characterization that is feasible for acquiring in practical applications. The results in Figure 9 show that the Bayesian estimator outperforms the competitors in terms of achieved MSE.

## 5. Application of the Proposed Model to Real Data

Without the loss of generality, this research was conducted using pavement data collected within the road network of the Republic of Kazakhstan; in particular, 2468 km of flexible pavements along highways (Figure 10 and Table 2) were surveyed. The type, severity, and extension of the distresses were measured with the Pavemetrics^®^ Laser Crack Measurement System (LCMS^®^), considering subsections 100 m long for a total number of sample units equal to 24,680. It has to be noticed that the application of this pavement distress estimation via Signal on Graph processing could be extended to every road network whose distress data are known regardless of the surveying technique adopted. 

In order to be able to evaluate the performance of the proposed model, the available dataset was split into two subsets of measurement sections: a training set (calibration phase) and a test set (validation phase), in the ratio approximately 75:25. Specifically, data collected along the A22 national highway were chosen as the test set, because of its 596 km length for a total number of sample units (called “Section Number” in the abscissa axis of the diagrams) equal to 5960, which represent 25% of the total road network length. The other roads were used as the training set, belonging to the same road class type. The training set was used to calculate the correlation matrix related to the investigated metrics, while the test set was used after the training to evaluate the model performance on unseen data. For these data, the model proposed in this paper was particularized as follows.

### 5.1. Construction of the Graph of the Metrics

The adjacency matrix coefficients $a_{ij}^{\left(1\right)}\;i,j=0,\;\ldots P-1$ were calculated as the correlation coefficients between the *i*-th and *j*-th distress metric values. Figure 11a illustrates, in pseudocolor, the adjacency matrix coefficients $a_{ij}^{\left(1\right)}\;i,j=0,\;\ldots P-1\;$ of the graph of the metrics. To reduce the metric graph complexity, and hence the computational burden, the adjacency matrix was thresholded by setting to zero the coefficients lower than 0.1. The considered thresholded adjacency matrix is shown in Figure 11b.

From Figure 11, we recognize that the most correlated metrics are those (i) at indexes 15, 16, 17, corresponding to different levels of raveling (see Table 1), and (ii) at indexes 8, 11 and 9, 12, corresponding to medium and high levels of longitudinal and transverse cracking, respectively. A smaller correlation effect is found between metrics 2–4, related to alligator cracking, and he metrics 10–12, representing transverse cracking. 

With these positions, we illustrate in Figure 12 the graph of the metrics as estimated on the training data. The *i*-th node represents the *i*-th metric; the *i*-th node size is proportional to the node degree, which is computed as the sum of the weights of the node’s edges, namely $\sum_{j=0}^{P-1}a_{ij}$, i=0, …P−1. The degree allows for identifying relevant, highly correlated metrics. Let us observe that the correlation of some road surface defects (namely ID 1, 13, and 14) with other cracks may be underestimated since these metrics were detected on few surveyed road sections, corresponding to less than 0.1% of the total number of sections. Moreover, it is possible to observe from Figure 12 that the metrics, namely ID 1, 6, 13, 14, and 18, are not correlated between them.

### 5.2. Construction of the Path Graph on Adjacent Spatial Sections

According to the above-described path graph model, each measurement is linked to the previous and following ones. In practice, when computing nonlinearity, the values assumed by the metric in the spatially neighboring sections are substituted by the averages over the five previous and next inspected sections. This operation averages out possible noise affecting the measurement. 

In summary, to correct the generic value $y_{n}^{\left(i\right)},$ the neighboring values over the nearest spatial sections are considered, as well as the correlated metrics. This is shown in Figure 13, where the black square represents the current metric $y_{n}^{\left(i\right)}$, the blue squares represent the correlated metrics acquired at the same section, and the yellow ones represent the spatially neighboring measurements averaged out to compute the spatial neighborhood estimates. 

A further remark about the metrics’ ranges is in order. Although being correlated, the measurements corresponding to different metrics span different ranges. According to the information in Table 1, Figure 14 shows the maximum measured values for each of the metrics, as computed on the test dataset. Before processing the measurements by the nonlinear estimator, they are normalized with respect to their maximum value. This operation prevents numerical errors and simplifies the selection of the $\beta_{i},i=1,\ldots \nu$ parameters. The result of the metric normalization is shown in Figure 15, which represents in pseudocolors the PxM matrix of the measurements.

## 6. Experimental Results

In this section, we illustrate the application of the Bayesian estimator to real data metrics. Specifically, we assume that the metrics are affected by generic acquisition noise, and we prove that they are corrected by the estimator. The nonlinearity parameters *p*_0_ and *β*_0_ are assigned values of 0.1 and 0.8, respectively. Regarding the assignment of the *β*_0_ parameter value to 0.8, it reflects the hypothesis of good accuracy in the observed data. The value attributed to *p*_0_ is in accordance with the high degree of confidence in the LCMS technology in detecting defects. It should be noted that the proposed method could also be applied considering data collection methods with less accuracy than those of the LCMS, defining lower values of the parameter *β*_0_. Herein, we aim at assessing the performance of the model for various real pavement distresses. 

Firstly, in order to quantitatively assess the ability of the estimator to recover from possible acquisition noise, we reproduce the experimental conditions as in Section 4.1, and we evaluate the MSE achieved by the proposed estimator and its competitors. The results are reported in Figure 16. We recognize that in this simulated environment, the random fluctuations due to the additive noise are reduced up to 25% with respect to the competitors. Thereby, the estimator is able to correct pseudo-random fluctuations by exploiting correlation among different metrics as well as among adjacent spatial locations.

Given this initial performance assessment, we now analyze the effect of the Bayesian estimator acting on the LCMS raw data. To this aim, we apply the estimator to each measured value, apart from the distress known as rutting (#4, 5 and 6), which assumes binary standard values, namely 0 or 45 m^2^. For this binary metric, the optimal nonlinearity slightly differs from that in (7), and following the approach in [54], it can be demonstrated to take the form of a soft thresholding stage. This notwithstanding, the values for the rutting metric are taken into account to adjust the values of the metrics that are sufficiently correlated with it. 

In Figure 17, we report the road distress measurements $y_{n}^{\left(i\right)}$ (blue dots), and the outputs of the Bayesian estimator ${\hat{x}}_{n}^{\left(i\right)}$ (orange dots). The single defects are represented. and the units of measurement are defined in Table 1; the measured values and the values replaced with the nonlinearity are compared: for the sake of clarity, a distress metric, namely the longitudinal cracking (Long.crk_H), is displayed in a larger scale. 

In order to make it easier to read the results, the difference between the measured starting values and the estimated values with the nonlinearity was also calculated section by section (Figure 18). We recognize that metrics characterized by large values, as for instance Long.crk_H, lead to larger numerical differences between the LCMS measured values and the estimated values with nonlinearity counterparts; this explains the larger differences spotted in Figure 18 for this metric. In summary, it is easy to detect the sections where the substitution of the values applying the proposed model has generated remarkable differences. The problem of identifying sections where the 3D laser technology can make mistakes is a highly attractive problem: the approach discussed in this paper may thus be a solution. 

Finally, we discuss the effect of the metric at the output of the Bayesian estimator on the PCI index. It should be noted that a numerical variation of the value of the single defect turns nonlinearly into the evaluation of the PCI index. For the above-described measurements, the PCI has been calculated with the implementation of a Visual Basic for Application (VBA) language-based program with interpolation by the parametric cubic spline of all of the density/deduct value curves of ASTM D6433 distress types [58], both in the case of the values measured with the LCMS and in the case of the values at the output of the Bayesian estimator. In particular, the variation of the PCI between the measurements and their processed version was statistically analyzed. Figure 19a,b plot the histogram and the cumulative density function of the PCI correction due to the adoption of the Bayesian estimator. As shown in Figure 19, the Bayesian estimator modifies the estimated PCI, and there are drastic changes in the estimated PCI values in only a few sections, where indeed the proposed signal on graph approach improves the quality of the LCMS measurements. Moreover, it can be observed that approximately for the 80% of the sections a lower PCI is returned by adopting the nonlinear approach: this result is in accordance with the study by Mulry et al. [39], which shows how the manually measured PCI is lower than the LCMS PCI.

## 7. Conclusions and Future Works

In this paper, we have introduced a novel Signal on Graph model of pavement distress metrics, by associating the values of the distress metrics acquired at different spatial sections to the vertices of a graph. The graph represents the correlation between different distress metrics at a given spatial section and the correlation among spatially adjacent measurements of the same metric. Then, we have derived a novel Bayesian estimator of the distress metrics, built on top of a recent model for signal defined on graphs. The estimator does not require large training annotated dataset, as linear and nonlinear regression techniques, since it leverages few parameters (mean, standard deviations) straightforwardly estimated from the observed data. Without loss of generality, we have assessed the methodology performance on a wide dataset of distress values, detected in some field tests carried out in Kazakhstan. The results can be summarized as follows:The related Bayesian estimator is effective in recovering acquisition errors, achieving an overall reduction of the error mean square value of about 25%, with respect to the considered competitors;The proposed methodology can be employed to identify sections where the measurements acquired by the 3D laser technology are unreliable related either to the limitations of the automated data collection or data processing software, as extensively discussed in the Introduction section. Indeed, a failure in the measurement system could be revealed when a large difference is computed at the same section between the observed value and the restored one obtained by the Bayesian estimator, i.e., the likelihood of the measured value is small;The proposed signal on graph approach selectively improves the quality of the LCMS measurements, as the Bayesian approach achieves a better accuracy in the PCI estimation compared to the LCMS PCI.

The herein presented signal on graph model of distress metrics can be applied independently on the acquisition method and it is viable for important developments. Firstly, the model naturally extends to account for side graph node information, such as scheduled road section maintenance or historical records. The graph-based structure also seamlessly encompasses Geographic Information System (GIS). Furthermore, the Bayesian estimator based on the signal on graph model of the distress measurements can be employed as a robust feature extraction stage at the input of a Deep Learning based pavement distress estimation system, leading to robust and accurate data-driven PCI estimation.

## Figures and Tables

**Figure 1 sensors-22-09183-f001:**
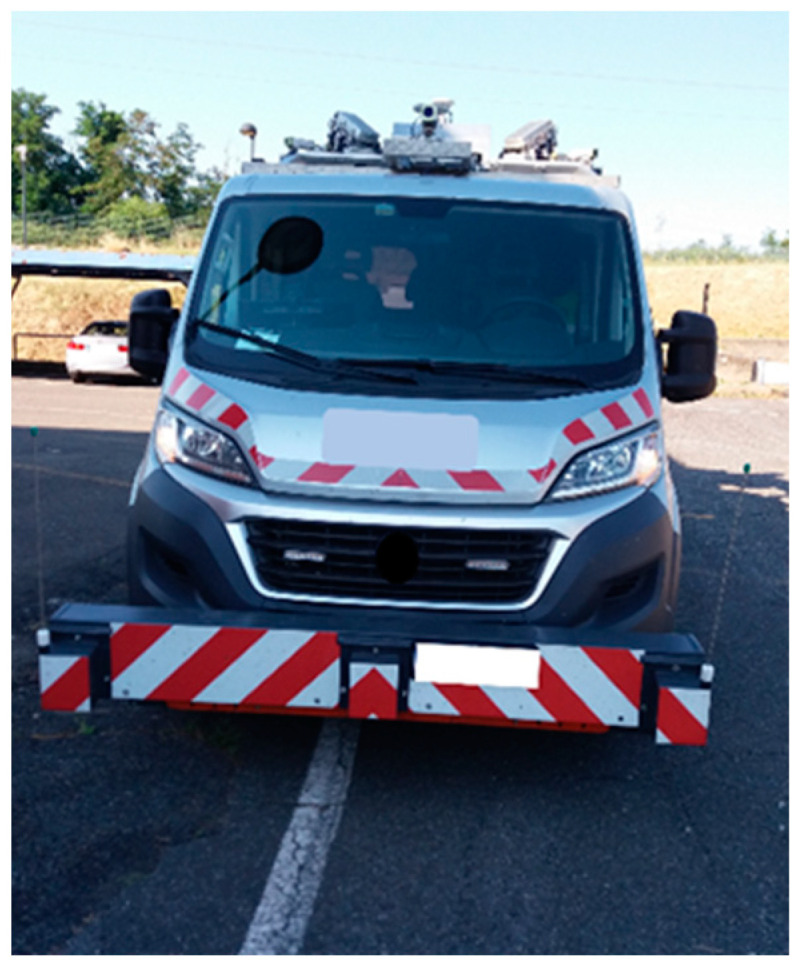
A modern data collection vehicle.

**Figure 2 sensors-22-09183-f002:**
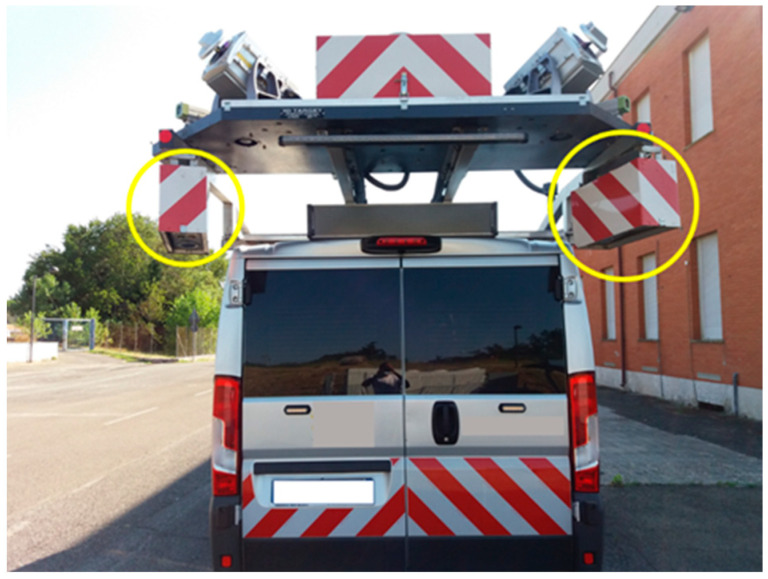
LCMS sensors installed on the surveying vehicle.

**Figure 3 sensors-22-09183-f003:**
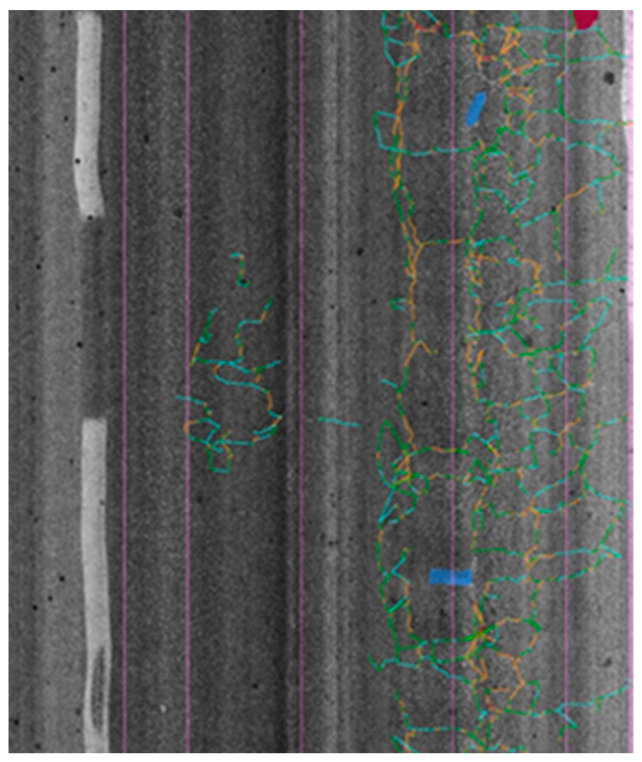
Pavement distress detected (severity = color code).

**Figure 4 sensors-22-09183-f004:**
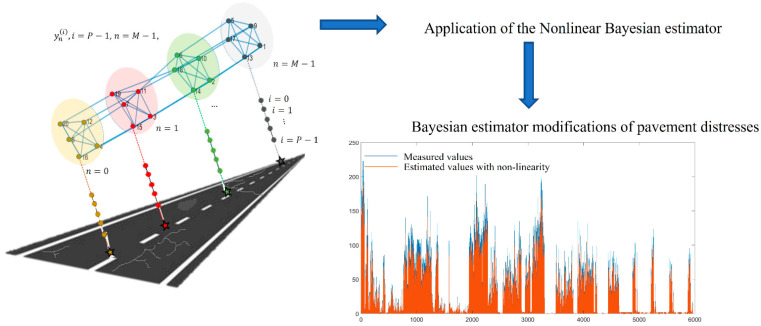
Framework for the pavement distress estimation via SoG processing.

**Figure 5 sensors-22-09183-f005:**
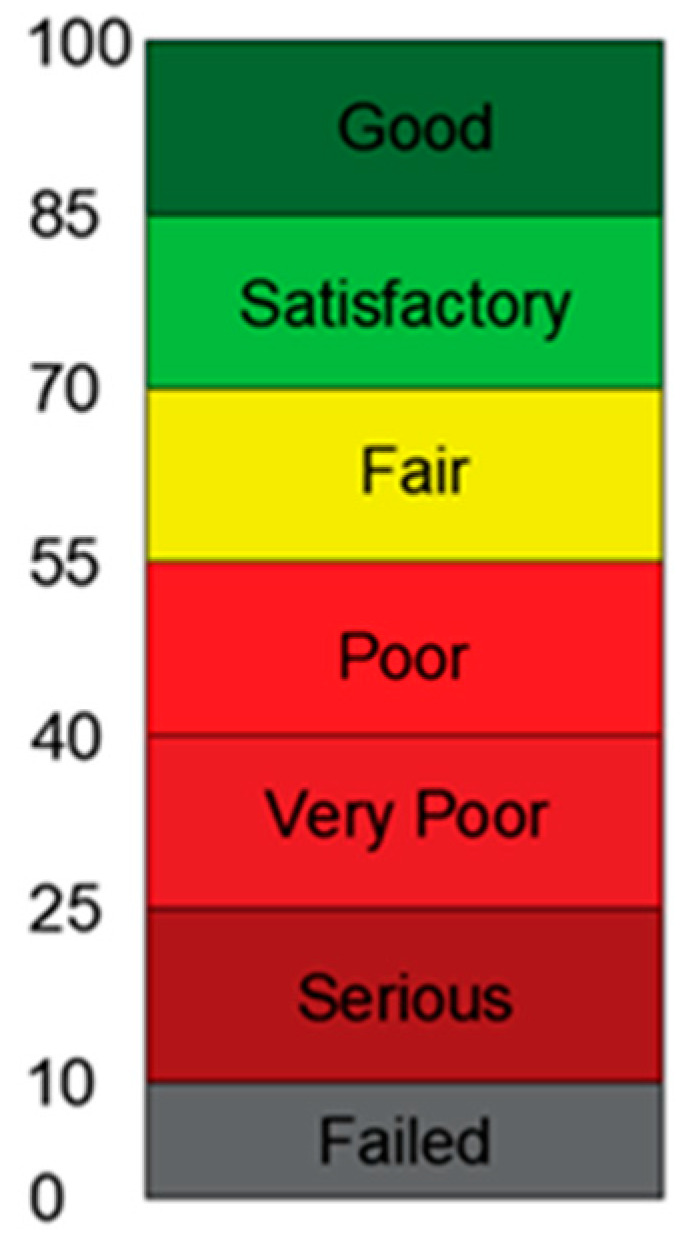
Standard PCI rating scale.

**Figure 6 sensors-22-09183-f006:**
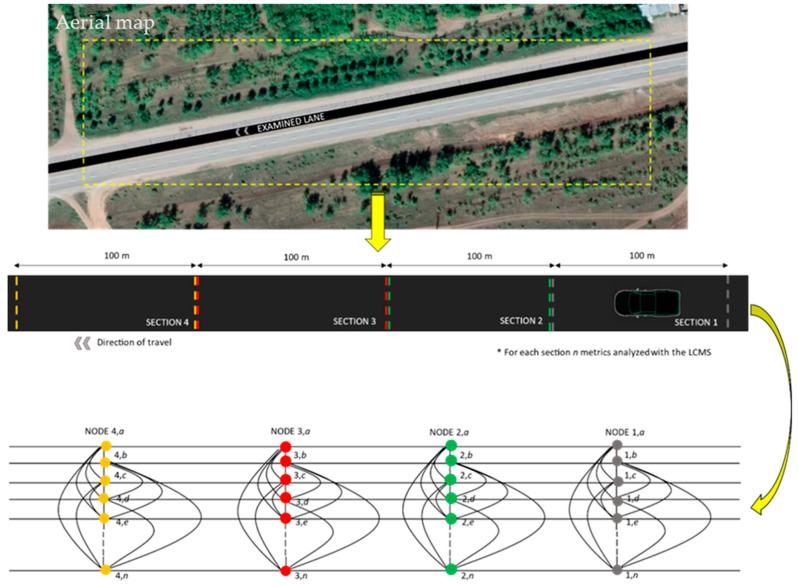
Distress road model as a signal on graph.

**Figure 7 sensors-22-09183-f007:**
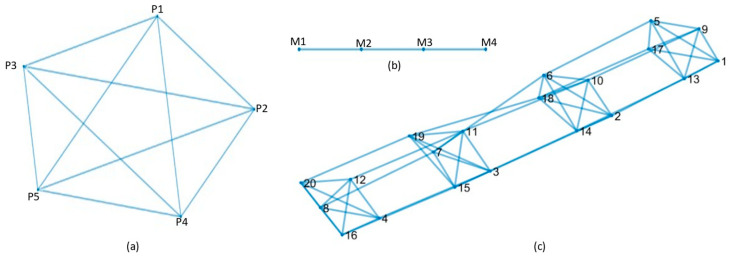
Product of graphs example for the toy case considering P = 5 metrics (P1, P2, P3, P4, P5) measured at M = 4 sections (M1, M2, M3, M4): (**a**) graph of the metrics’ correlation at a given section); (**b**) path graph representing the similarity of the same metric over adjacent spatial sections; (**c**) overall product graph.

**Figure 8 sensors-22-09183-f008:**
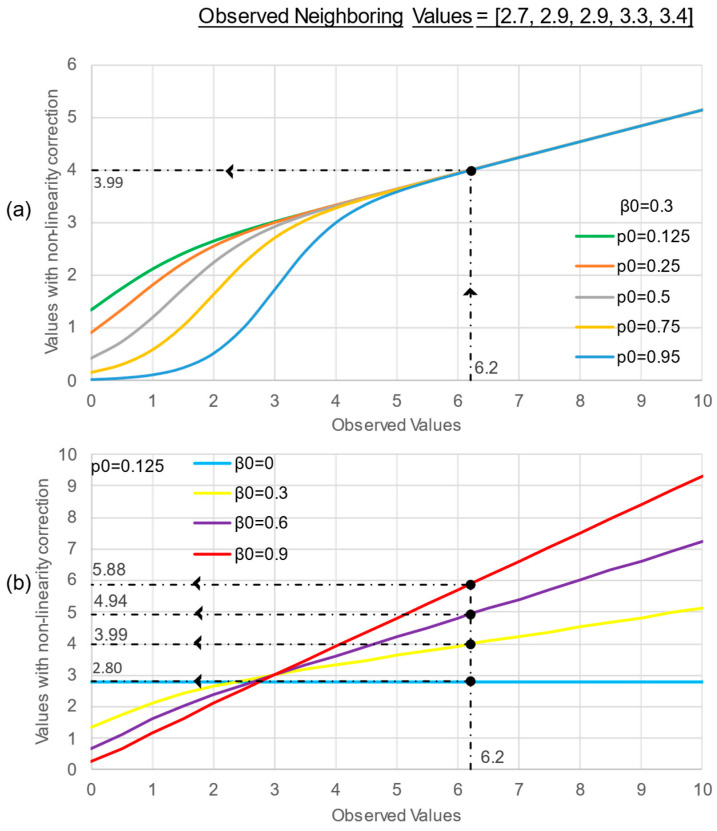
Effect of the nonlinear estimator on the observed value as the *p*_0_ (**a**) and the *β*_0_ parameter (**b**) change; the case of observed value equal to 6.2 is highlighted.

**Figure 9 sensors-22-09183-f009:**
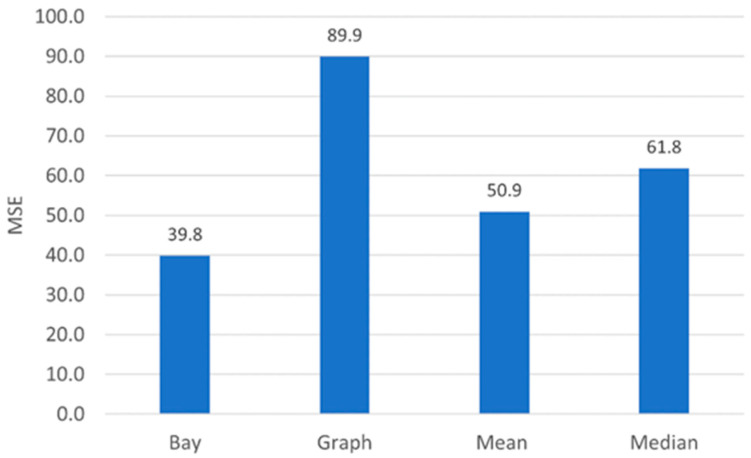
P = 5 metrics, M = 4 sections graph schematically represented in Figure 6. MSE of the Bayesian estimator (Bay), the linear estimate obtained averaging over neighboring graph vertices (Graph), the linear estimate obtained averaging over spatially adjacent vertices (Mean) and the nonlinear estimate obtained as the median over spatially adjacent vertices (Median).

**Figure 10 sensors-22-09183-f010:**
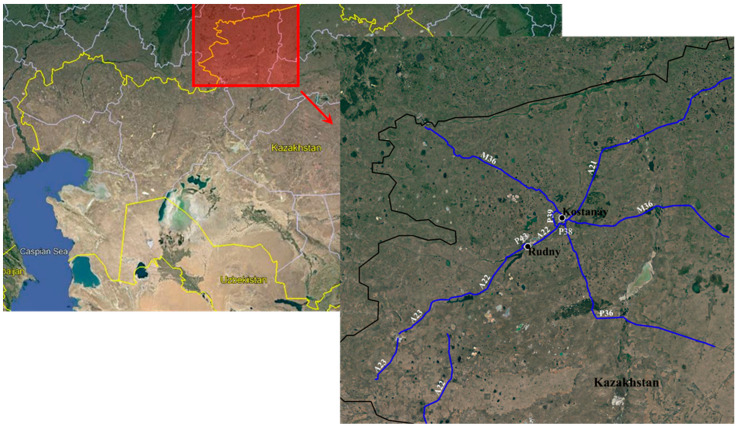
Examined roads—2468 km.

**Figure 11 sensors-22-09183-f011:**
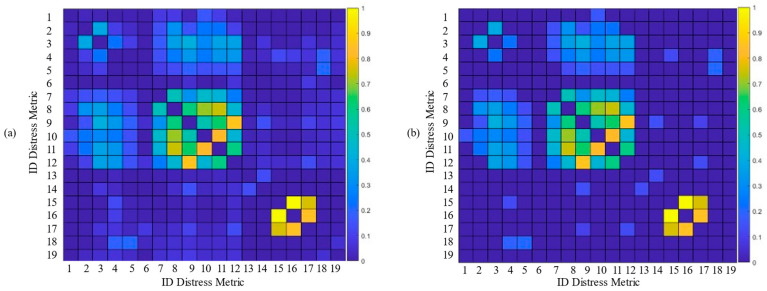
Adjacency matrix coefficients $a_{ij}^{\left(1\right)}\;i,j=0,\;\ldots P-1$ of the graph of the metrics, calculated as the correlation coefficients between the *i*-th and *j*-th distress metric values: (**a**) original values and (**b**) thresholded values with threshold ϑ=0.1.

**Figure 12 sensors-22-09183-f012:**
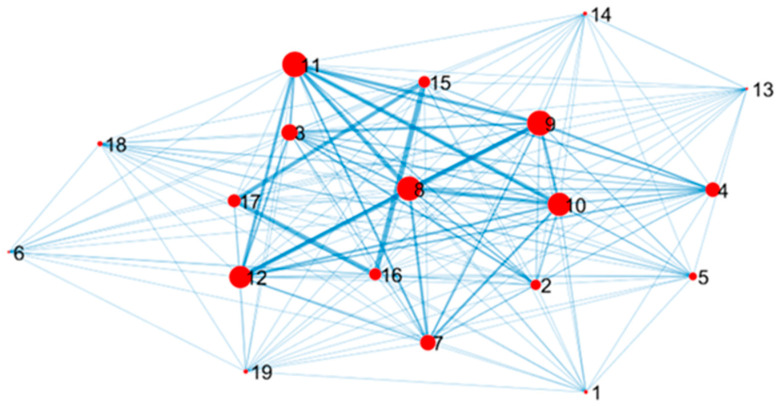
Correlations between the different investigated metrics.

**Figure 13 sensors-22-09183-f013:**
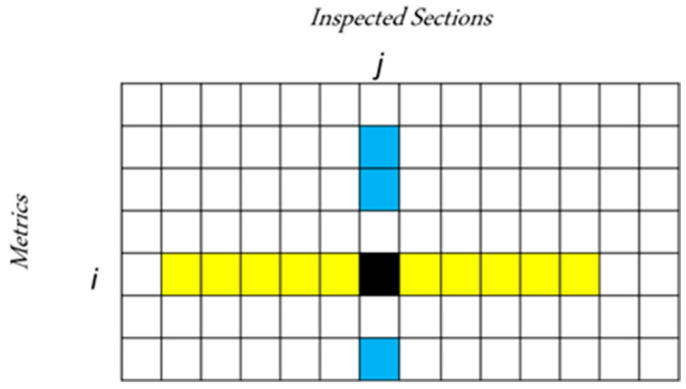
Measurements involved in the estimation of the *i*-th metric at the n-th inspected section (black square): measurements relative to correlated metrics at the same inspected section (light blue squares), and measurements of the same metric averaged over spatially adjacent sections (yellow squares).

**Figure 14 sensors-22-09183-f014:**
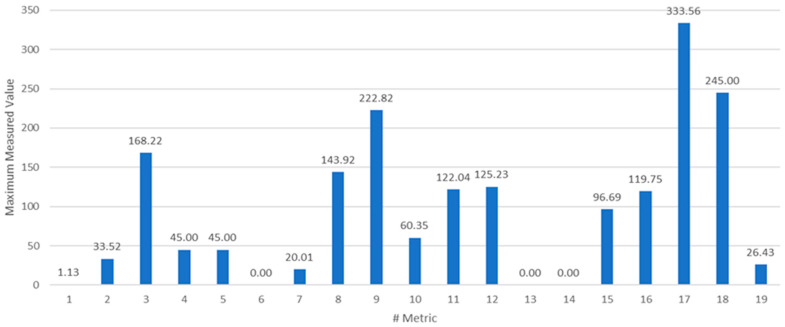
The maximum measured values for each of the metrics.

**Figure 15 sensors-22-09183-f015:**
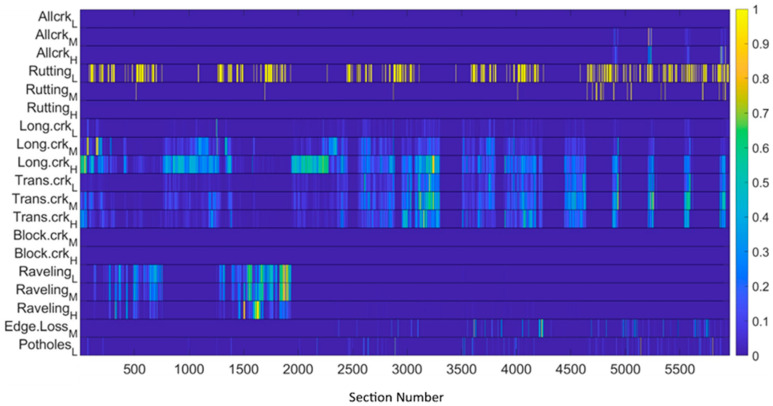
Metrics normalized to the maximum measured values.

**Figure 16 sensors-22-09183-f016:**
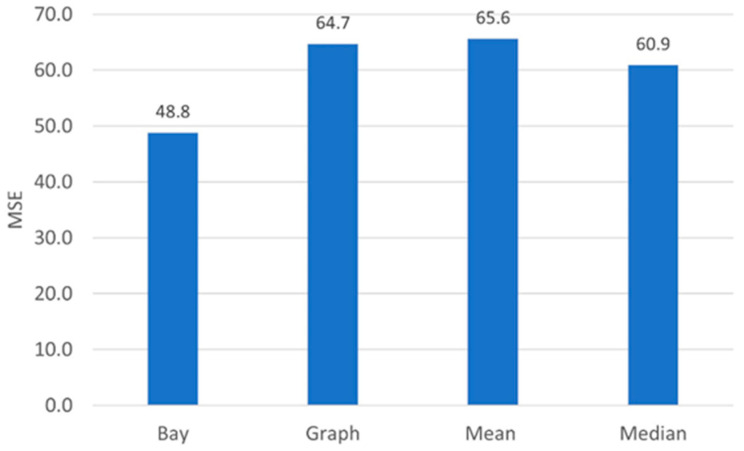
Test Set. MSE of the Bayesian estimator (Bay), the linear estimate obtained averaging over neighboring graph vertices (Graph), the linear estimate obtained averaging over spatially adjacent vertices (Mean) and the nonlinear estimate obtained as the median over spatially adjacent vertices (Median).

**Figure 17 sensors-22-09183-f017:**
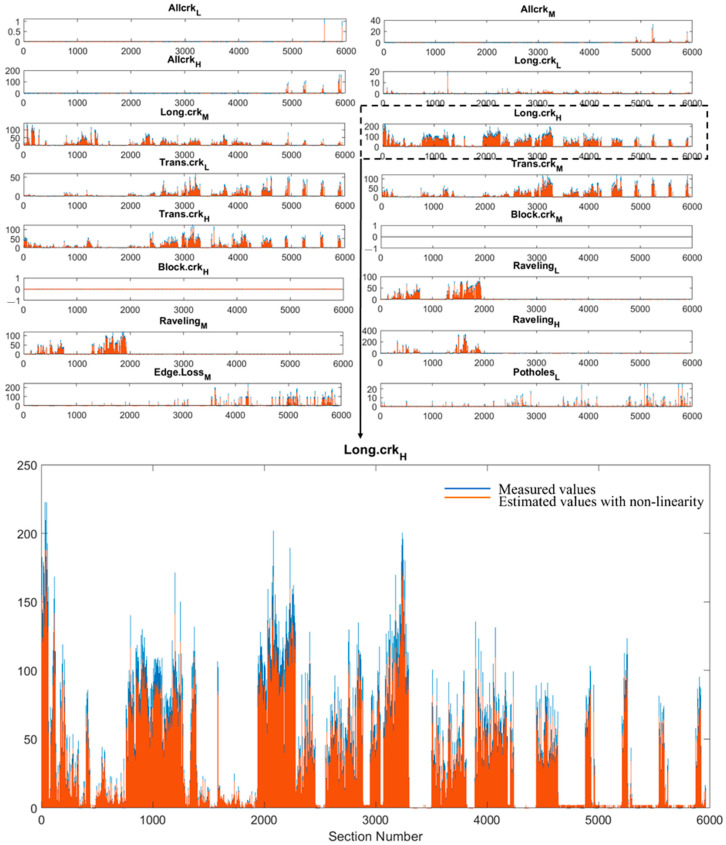
The measured values (blue) and the estimated values with nonlinearity (orange) for each of the metrics.

**Figure 18 sensors-22-09183-f018:**
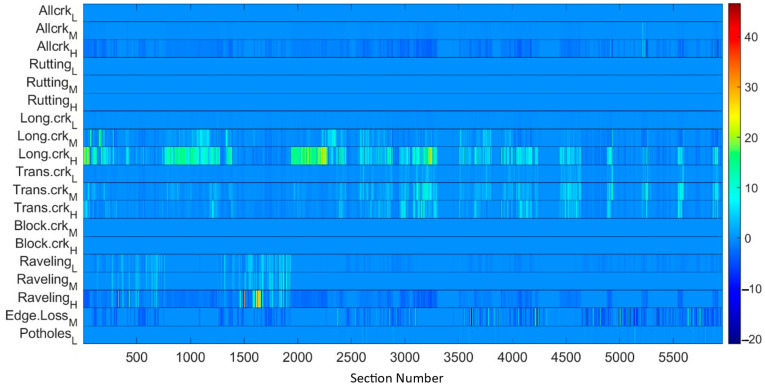
The difference between the measured values and the estimated values with non-linearity.

**Figure 19 sensors-22-09183-f019:**
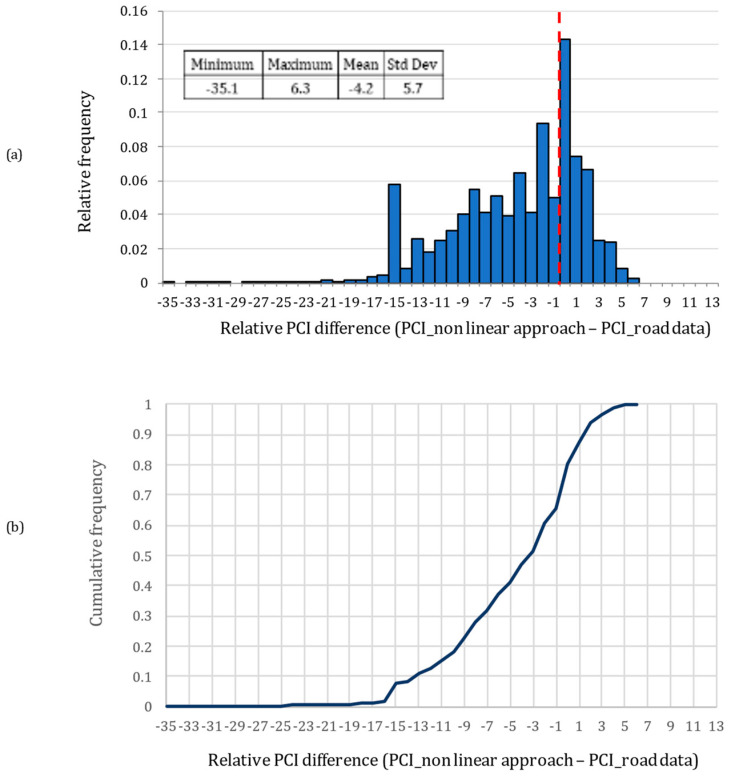
PCI correction due to Bayesian estimator in terms of relative frequency (**a**) and cumulative frequency (**b**).

**Table 1 sensors-22-09183-t001:** Road surface defects considered in the development of the model proposed in this study.

ID	ASTM Name	Degree of Severity	Unit of Measure	Cause
1	Alligator cracking	Low	m^2^	Load
2		Medium	m^2^	Load
3		High	m^2^	Load
4	Rutting	Low	m^2^	Load
5		Medium	m^2^	Load
6		High	m^2^	Load
7	Longitudinal cracking	Low	m	Climatic/Construction defect
8		Medium	m	Climatic/Construction defect
9		High	m	Climatic/Construction defect
10	Transverse cracking	Low	m	Climatic/Construction defect
11		Medium	m	Climatic/Construction defect
12		High	m	Climatic/Construction defect
13	Block cracking	Medium	m^2^	Climatic
14		High	m^2^	Climatic
15	Raveling	Low	m^2^	Bituminous mixture low quality
16		Medium	m^2^	Bituminous mixture low quality
17		High	m^2^	Bituminous mixture low quality
18	Lane/Shoulder Drop-off	Medium	m	Climatic/Poor construction
19	Potholes	Low	[-]	Traffic, load

**Table 2 sensors-22-09183-t002:** Road surface defects considered in the development of the model proposed in this study.

Road Name	Category	Direction
M36	International Highway	Border of Russia—Kostanay—Astana—Karaganda—Almaty
A21	National Highway	Mamlyutka—Kostanay
A22	National Highway	Karabutak—Komsomol’skoe, Kazakhstan—Denisovka—Rudny, Kazakhstan—Kostanay
A23	National Highway	Denisovka—Zhetikara—Muktikol
P36	Regional Highway	Kostanay—Auliekol—Surgan
P38	Regional Highway	Kostanay Southern Bypass (At Minchurinskoe)
P39	Regional Highway	Kostanay Western Bypass
P43	Regional Highway	Rudny West Bypass

## Data Availability

The data presented in this study are available on request from the corresponding author. The data are not publicly available due to confidentiality reasons.

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
