# Peer review of "Pavement Distress Estimation via Signal on Graph Processing"

_sensors, 2022, doi:10.3390/s22239183_

Round 1

Reviewer 1 Report

This paper presents a new technique for improving automatic pavement distress detection systems. The manuscript is well-written and attractive, and it only needs minor improvement before it can be recommended for publication.

(1) Try to merge section 2 into 1.

(2) The title of Figure 6 is too long.

(3) Please given more details to clarify the statement: “The proposed methodology can be employed to identify sections where the measurement acquired by the 3D laser technology is unreliable.”

Author Response

This paper presents a new technique for improving automatic pavement distress detection systems. The manuscript is well-written and attractive, and it only needs minor improvement before it can be recommended for publication.

We thank the Reviewer for the positive feedback and fruitful comments.

  • Try to merge section 2 into 1.

We thank the Reviewer for the suggestion. We have merged the 2 sections and we have introduced two subsections in the Introduction, the “Background” and the “Objectives of the current work and problem statement”, in particular.

  • The title of Figure 6 is too long.

We thank the Reviewer for the suggestion. We have shortened the title length and explained the figure into the text.

  • Please given more details to clarify the statement: “The proposed methodology can be employed to identify sections where the measurement acquired by the 3D laser technology is unreliable.”

We thank the Reviewer for the comment which allows us to clarify this point. We have modified the statement as follow: “The proposed methodology can be employed to identify sections where the measurements acquired by the 3D laser technology are unreliable related either to the limitations of the automated data collection or data processing software, as extensively discussed in the Introduction section. Indeed, a failure in the measurement system could be revealed when a large difference is computed at the same section between the observed value and the restored one obtained by the Bayesian estimator, i.e. the likelihood of the measured value is small.” See lines 577-583.

Reviewer 2 Report

This paper gives a method to dectect and evaluate pavement performance. Based on the pavement detect and  recognize processing, the results can be used to pavement performance evaluation. But the crack width may chang from 0.1mm to 3-5mm, can you give some difination and some limitation for those crack types.

Author Response

This paper gives a method to dectect and evaluate pavement performance. Based on the pavement detect and recognize processing, the results can be used to pavement performance evaluation. But the crack width may chang from 0.1mm to 3-5mm, can you give some difination and some limitation for those crack types

We thank the Reviewer for the positive feedback and fruitful comment, which allows us to clarify this point. We apologise for not giving the definition of the longitudinal and transverse cracking in the paper, as well as the rutting one. We have added these sentences at lines 194-201: “Rutting is a road surface depression or groove due to the travel of the wheels or skis. The deformation of the asphalt concrete pavement or subbase material can cause the ruts, which can become unsafe during rainy events, when the paths are filled with water. Longitudinal and transverse cracking happens lengthwise or crosswise at approximately right angles to the pavement’s centerline or laydown direction. These types of crack are not usually load-associated, but can occur, for example, at low temperature or because of asphalt hardening. Nonfilled crack widths less than 6 mm, for airport pavement, and 10 mm, for road pavement, are always considered of low severity.”
